# Deep Learning-Based Screening of Urothelial Carcinoma in Whole Slide Images of Liquid-Based Cytology Urine Specimens

**DOI:** 10.3390/cancers15010226

**Published:** 2022-12-30

**Authors:** Masayuki Tsuneki, Makoto Abe, Fahdi Kanavati

**Affiliations:** 1Medmain Research, Medmain Inc., Fukuoka 810-0042, Japan; fkanavati@medmain.com; 2Department of Pathology, Tochigi Cancer Center, 4-9-13 Yohnan, Utsunomiya 320-0834, Japan; makotabe@tochigi-cc.jp

**Keywords:** urothelial carcinoma, urine, liquid-based cytology, deep learning, cancer screening, whole slide image

## Abstract

**Simple Summary:**

In this study, we aimed to investigate the use of deep learning for classifying whole-slide images of urine liquid-based cytology specimens into neoplastic and non-neoplastic (negative). To do so, we used a total of 786 whole-slide images to train models using four different approaches, and we evaluated them on 750 whole-slide images. The best model achieved good classification performance, demonstrating the promising potential of use of such models for aiding the screening process for urothelial carcinoma in routine clinical practices.

**Abstract:**

Urinary cytology is a useful, essential diagnostic method in routine urological clinical practice. Liquid-based cytology (LBC) for urothelial carcinoma screening is commonly used in the routine clinical cytodiagnosis because of its high cellular yields. Since conventional screening processes by cytoscreeners and cytopathologists using microscopes is limited in terms of human resources, it is important to integrate new deep learning methods that can automatically and rapidly diagnose a large amount of specimens without delay. The goal of this study was to investigate the use of deep learning models for the classification of urine LBC whole-slide images (WSIs) into neoplastic and non-neoplastic (negative). We trained deep learning models using 786 WSIs by transfer learning, fully supervised, and weakly supervised learning approaches. We evaluated the trained models on two test sets, one of which was representative of the clinical distribution of neoplastic cases, with a combined total of 750 WSIs, achieving an area under the curve for diagnosis in the range of 0.984–0.990 by the best model, demonstrating the promising potential use of our model for aiding urine cytodiagnostic processes.

## 1. Introduction

For routine clinical practices, clinicians obtain urinary tract cytology specimens for the screening of urothelial carcinoma [1,2]. Urine specimens play a critical role in the clinical evaluation of patients who have clinical signs and symptoms (e.g., haematuria and painful urination) suggestive of pathological changes within the urinary tract [3]. Urothelial carcinoma is the most common malignant neoplasm detected by urine cytology. The most common site of origin of urothelial carcinoma is bladder. According to the Global Cancer Statistics 2020 [4], bladder cancer is the tenth most commonly diagnosed cancer with 573,278 of new cases and 212,536 of new deaths worldwide in 2020. Most of the bladder cancers are urothelial in origin (approximately 90% of bladder cancers) and primary adenocarcinoma of the bladder is rare [3,5,6]. Bladder cancer often presents insidiously. Haematuria is the most common presentation of bladder cancer, which is typically intermittent, frank, painless and at times present throughout micturition [3]. Delayed diagnosis of urothelial carcinoma is associated with high grade muscle invasion which has the potential to progress rapidly and cancer metastasis [3]. Of course, cystoscopy with a biopsy is the gold standard for diagnosis of urothelial carcinoma in clinical practice; however, it is aggressive and relatively inconvenient as a follow-up monitoring approach [7]. It has been reported that 48.6% of biopsy proven low-grade urothelial carcinomas had a urine cytodiagnosis of atypical or neoplastic suspicious, which could conclude that existing urine cytology screening and surveillance systems are accurate in diagnosing urothelial carcinoma [8]. Therefore, cytological urothelial carcinoma screening in urine specimens plays a key role in early stage cancer detection and treatment in routine clinical practices [9,10].

Liquid-based cytology (LBC) was developed as an alternative to conventional smear cytology in the 1990s [11]. LBC has several advantages in preparation and diagnostic process compared with conventional smear [12,13,14,15]. The LBC technique preserves the cells of interest in a liquid medium and removes most of the debris, blood, and exudate either by filtering or density gradient centrifugation [16,17]. LBC provides automated and standardized processing techniques that produce a uniformly distributed and cell-enriched slide [18,19,20]. Moreover, residual specimens can be used for additional investigations (e.g., immunocytochemistry) [21,22,23]. ThinPrep (Hologic, Inc., Marlborough, MA, USA) and SurePath (Becton Dickinson, Inc., Franklin Lakes, NJ, USA) for LBC specimen preparation have been approved by the US Food and Drug Administration (FDA). Compared to the conventional smear cytology, LBC has lower background elements, provides better cell preservation, and has a higher satisfaction rate [24]. As for the sensitivity, it has been reported that LBC achieved at 0.58 (CI: 0.51–0.65) and conventional smear achieved at 0.38 [7,11]. It is understandable that the efficiency of diagnosis employing the LBC is high because of the cell collection rate. It was shown that the accuracy of diagnoses made employing the LBC method can be increased by understanding the characteristics of the cell morphology in suspicious cases (e.g., high-grade urothelial carcinoma and low-grade urothelial carcinoma [25], and in other malignancies [26,27]). LBC specimens performed significantly better in urinary cytology when evaluating malignant categories especially high-grade urothelial carcinoma (HGUC), which facilitate a more accurate diagnosis than conventional preparations [15]. Moreover, from the standpoint of rationality, preparation and screening times were 2.25 and 1.33–2.00 times greater when using LBC (ThinPrep) compared with cytocentrifugation (conventional smear cytology) [28]. Therefore, computational screening (cytodiagnostic) aids for urine LBC specimens would be a great benefit for urothelial carcinoma screening as medial image analysis.

Whole-slide images (WSIs) are digitisations of the conventional glass slides obtained via specialised scanning devices (WSI scanners), and they are considered to be comparable to microscopy for primary diagnosis [29]. It has been reported that evaluation of WSI is generally equivalent to using conventional glass slides under microscopy [30]. The use of WSI has to some degree met the goals of saving pathologists working time and providing high quality pathological images with convenient access and easily navigable viewing online based software which saves resources and costs by eliminating slide glass shipping expenses [30]. The advent of WSIs led to the application of medical image analysis techniques, machine learning, and deep learning techniques for aiding pathologists in inspecting WSIs [31]. Importantly, a routine scanning of LBC slides in a single layer of WSIs would be suitable for further high throughput analysis (e.g., automated image based cytological screening and medical image analysis) [20]. Indeed, deep learning approaches and its clinical application to classify cytopathological changes (e.g., neoplastic transformation) were reported in the recent years [32,33,34,35,36,37,38,39,40,41].

In this study, we trained deep learning models based on convolutional neural networks (CNN) using a training dataset of 786 urine LBC (SurePath) WSIs. We evaluated the model on two test sets with a combined total of 750 WSIs, achieving ROC area under the curve (AUC) for WSI neoplastic classification in the range of 0.984–0.990.

## 2. Materials and Methods

### 2.1. Clinical Cases and Cytopathological Records

In this retrospective study, a total of 1556 LBC SurePath (Becton Dickinson) conventionally prepared cytopathological slide glass specimens of human urine cytology were collected from a private clinical laboratory in Japan after routine cytopathological review of those glass slides by cytoscreeners and pathologists. The private clinical laboratory in Japan that provided urine LBC specimen glass slides in the present study was anonymized due to the confidentiality agreement. The LBC specimens were selected randomly to reflect a real clinical settings as much as possible. We have also collected LBC specimens so as to compile test sets with an equal balance and a clinical balance of negative and neoplastic. The equal balance test set consisted of 50% negative and 50% neoplastic urine LBC cases (Table 1). The clinical balance test set consisted of a ratio of 10 (negative) to 1 (neoplastic) urine LBC cases based on a real clinical setting which was reported by the Japanese Society of Clinical Cytology as the statistics on cytodiagnosis in 2016 to 2021 (https://jscc.or.jp/, accessed on 27 January 2022). Prior to the start of the experiments, the cytoscreeners and pathologists excluded inadequate LBC specimens (*n* = 21) which had inadequate cellularity or had significant artifacts like dust or ink markings. All WSIs were scanned at a magnification of ×20 using the same Leica Aperio AT2 Digital Whole Slide Scanner (Leica Biosystems, Tokyo, Japan) and were saved in the SVS file format with JPEG2000 compression. Each WSI was observed by at least two cytoscreeners or pathologists to confirm the diagnosis, with the final checking and verification performed by a senior cytoscreener or pathologist. We have confirmed that cytoscreeners and pathologists were able to classify (Table 1) from the visual inspection of the LBC SurePath (Becton Dickinson) stained WSIs alone.

In this study, we have classified urine LBC WSIs into two classes: one is negative and the other is neoplastic (Table 1). Negative WSIs were diagnosed as Class I or Class II and neoplastic WSIs were diagnosed as Class III, Class IV, or Class V in routine clinical cytodiagnosis (Table 1) (Class I: negative for HGUC; Class II: negative for HGUC with reactive urothelial epithelial cells; Class III: atypical urothelial epithelial cells and suspicious for LGUC; Class IV: LGUC and suspicious for HGUC; Class V: HGUC). The cytoscreeners and pathologists had to agree whether the output class was negative or neoplastic on each urine LBC WSI.

### 2.2. Annotation

A cohort of 62 training cases and 10 validation cases were manually annotated by experienced pathologists (Table 1). Coarse manually drawing polygonal annotations were obtained by free-hand drawing in-house online tool developed by customising the open-source OpenSeadragon tool at https://openseadragon.github.io/ (accessed on 25 July 2021) which is a web-based viewer for zoomable images. On average, the cytoscreeners and pathologists manually annotated 180 cells (or cellular clusters) per WSI. Annotated neoplastic WSIs consisted of Class III, Class IV, and Class V cytodiagnostic classes except for Class I and Class II (Table 1). We set three annotation labels for neoplastic urothelial epithelial cells: atypical cell, low-grade urothelial carcinoma (LGUC) cell, and high-grade urothelial carcinoma (HGUC) cell (Table 2 and Figure 1). For example, on the Class III (Figure 1A,B), Class IV (Figure 1C,D), and Class V (Figure 1E,F) WSIs, cytoscreeners and pathologists performed annotations around the atypical cells (Figure 1A,B), LGUC cells (Figure 1C,D), and HGUC cells (Figure 1E,F) based on the representative neoplastic urothelial epithelial cell morphology (e.g., hyperchromatism, irregular chromatin distribution, abnormalities of nuclear shape, increased nuclear/cytoplasmic ratio, irregular nuclear distribution, nuclear enlargement, abnormal cytoplasm, prominent nucleolus, cellular and nuclear polymorphism). If the WSIs were classified as Class V, for example, it would be possible to have atypical cell, LGUC cell, and HGUC cell annotations in a WSI. In contrast, the cytoscreeners and pathologists did not annotate areas where it was difficult to cytologically determine that the cells were neoplastic. The negative subset of the training and validation sets (Table 1) was not annotated and the entire cell spreading areas within the WSIs were used. The average annotation time per WSI was about 90 min. Annotations performed by the cytoscreeners and pathologists were modified (if necessary) and verified by a senior cytoscreener.

### 2.3. Deep Learning Models

We performed training using transfer learning with fine-tuning using two different weight initialisations: ImageNet (IN) and pre-training on a uterine cervix (UC) neoplastic (×10, 1024) dataset from a previous study [35]. We used two different approaches for training during fine-tuning: fully supervised (FS) and weakly supervised (WS) learning. We used a modified version of EfficientNetB1 (ENB1) [42] with a tile size of 1024 × 1024 px. This resulted in a total of four models, all trained at magnification ×10 and tile size 1024 × 1024 px: ENB1-UC-FS+WS, ENB1-UC-WS, ENB1-IN-FS+WS, and ENB1-IN-WS. In addition, for comparaison with other model architectures, we trained models using ResNet50V2 [43], DenseNet121 [44] and InceptionV3 [45]. For these models we trained uisng FS + WS method and with initialisation from ImageNet, as we did not have access to models trained with these architecture on uterine cervix. We performed the fine-tuning of the models using the partial fine-tuning approach [46], which consists of only fine-tuning the affine parameters of batch-normalization layers and the final classification layer (Figure 2). starting with pre-trained weights on ImageNet.

Figure 2 shows an overview of the training method and trained deep learning models. The training methodology that we used in the present study was exactly the same as reported in our previous studies [47].

We performed slide tiling by extracting square tiles from tissue regions of the WSIs. We started by detecting the tissue regions in order to eliminate most of the white background. This was conducted by performing thresholding on a grayscale version of the WSIs using Otsu’s method [48]. For the CNN, we have used the EfficientNetB1 architecture [42] with a modified input size of 1024 × 1024 px to allow a larger view; this is based on cytologists’ input that they usually need to view the neighbouring cells around a given cell in order to diagnose more accurately. We used the partial fine-tuning approach [46] for the tuning the CNN component.

For training and inference, we then proceeded by extracting 1024 × 1024 px tiles from the tissue regions. We performed the extraction in real-time using the OpenSlide library [49]. To perform inference on a WSI, we used a sliding window approach with a fixed-size stride of 512 × 512 px (half the tile size). This results in a grid-like output of predictions on all areas that contained cells, which then allowed us to visualise the prediction as a heatmap of probabilities that we can directly superimpose on top of the WSI. Each tile had a probability of being neoplastic; to obtain a single probability that is representative of the WSI, we computed the maximum probability from all the tiles.

During fully supervised learning, we maintained an equal balance of positively and negatively labelled tiles in the training batch. To do so, for the positive tiles, we extracted them randomly from the annotated regions (annotation label: atypical cell, LGUC cell, and HGUC cell) of neoplastic WSIs, such that within the 1024 × 1024 px, at least one annotated cell was visible anywhere inside the tile. For the negative tiles, we extracted them randomly anywhere from the tissue regions of negative WSIs (Table 1). We then interleaved the positive and negative tiles to construct an equally balanced batch that was then fed as input to the CNN. In addition, to reduce the number of false positives, given the large size of the WSIs, we performed a hard mining of tiles, whereby at the end of each epoch, we performed full sliding window inference on all the negative WSIs in order to adjust the random sampling probability such that false positively predicted tiles of negative were more likely to be sampled.

During weakly supervised learning, to maintain the balance on the WSI, we oversampled from WSIs to ensure that the model trained on tiles from all WSIs in each epoch. We then switched to hard mining tiles. To perform hard mining, we alternated between training and inference. During inference, the CNN was applied in a sliding window fashion on all the tissue regions in the WSI, and we then selected the *k* tiles with the highest probability for being positive. This step effectively selects tiles that are most likely to be false positives when the WSI is negative. The selected tiles were placed in a training subset, and once that subset contained *N* tiles, training was initiated. We used 
k=8
, 
N=256
, and a batch size of 32.

During training, we performed real-time augmentation of the extracted tiles using variations of brightness, saturation, and contrast. We trained the model using the Adam optimisation algorithm [50], with the binary cross entropy loss, 
$beta_{1}=0.9$
, 
$beta_{2}=0.999$
, and a learning rate of 
0.001
. We applied a learning rate decay of 
0.95
 every 2 epochs. We used early stopping by tracking the performance of the model on a validation set, and training was stopped automatically when there was no further improvement on the validation loss for 10 epochs. The model with the lowest validation loss was chosen as the final model.

### 2.4. Software and Statistical Analysis

The deep learning models were implemented and trained using the open-source TensorFlow library [51]. AUCs were calculated in python using the scikit-learn package [52] and plotted using matplotlib [53]. The 95% CIs of the AUCs were estimated using the bootstrap method [54] with 1000 iterations. The ROC curve was computed by varying the probability threshold from 0.0 to 1.0 and computing both the TPR and FPR at the given threshold.

## 3. Results

### 3.1. Insufficient AUC Performance of Whole Slide Image (WSI) Neoplastic Evaluation on Urine LBC WSIs Using Existing Series of LBC Cytopathological Model

Prior to training urine LBC neoplastic screening models, we applied existing LBC uterine cervix neoplastic screening model [35] and histopathological classification models and evaluated their AUC performances on urine LBC test sets (Table 1). This is summarised in Table 3.

### 3.2. High ROC-AUC Performance of Urine LBC WSI Evaluation of Neoplastic Urothelial Epithelial Cell Screening

We trained four deep learning models ([ENB1-UC-FS+WS], [ENB1-UC-WS], [ENB1-IN-FS+WS], and [ENB1-IN-WS]) using transfer learning (TL) with fine-tuning [47] with fully supervised (FS) learning [35,55], and weakly supervised (WS) learning [56] approaches as described elsewhere. These models are all based on the EfficientNetB1 convolutional neural network (CNN) architecture. To compare transfer learning models’ performance ([ENB1-UC-FS+WS] and [ENB1-UC-WS]), we trained two models using EfficientNetB1 architecture at a same magnification of ×10 and tile size (1024 × 1024 px). To train deep learning models, we used a total of 62 neoplastic (with annotation) and 724 negative (without annotation) training set WSIs and 10 neoplastic (with annotation) and 10 negative (without annotation) validation set WSIs (Table 1). This resulted in four different models: (1) ENB1-UC-FS+WS (×10, 1024), (2) ENB1-UC-WS (×10, 1024), (3) ENB1-IN-FS+WS (×10, 1024), and (4) ENB1-IN-WS (×10, 1024). We evaluated these four different trained deep learning models on equal balance and clinical balance test sets (Table 1). For each test set (equal and clinical balance), we computed the ROC-AUC, log-loss, accuracy, sensitivity, and specificity and summarized in Table 4 and Figure 3 and Figure 4. Overall, four different trained deep learning models achieved equivalent ROC-AUC, log-loss, accuracy, sensitivity, and specificity at whole-slide level (WSI evaluation) in both equal and clinical balance test sets (Table 4, Figure 3). However, heatmap image appearances were different among four trained deep learning models (Figure 4). The localization patterns of predicted tiles were approximately same among four trained deep learning models (Figure 4). Looking at heatmap images of the same urine LBC WSIs (WSI-1 and WSI-2) (Figure 4) that were correctly predicted (true-positive) as neoplastic WSI using four different trained models, all models could predict tiles with neoplastic urothelial epithelial cells (Figure 4A–D,M–P) satisfactorily (Figure 4E–L,Q–X). However, probabilities in each neoplastic predicted tiles were totally different among four trained models (Figure 4). Among the four trained model, ENB1-UC-FS+WS exhibited the best tile prediction overall based on inspection of the heatmap images (Figure 4E,F,Q,R). Therefore, our results show that the ENB1-UC-FS+WS model is the best model for urine LBC neoplastic urothelial epithelial cell screening (Table 4 and Figure 4). Figures show representative WSIs of true positive (Figure 5), true negative (Figure 6), false positive (Figure 7), and false negative (Figure 8) from using the model [ENB1-UC-FS+WS].

### 3.3. True Positive Prediction

The model ENB1-UC-FS+WS satisfactorily predicted neoplastic urothelial epithelial cells in urine LBC WSIs (Figure 5). Cytopathologically, Figure 5A exhibited atypical urothelial epithelial cells (Figure 5B) and was diagnosed as Class III. Figure 5E showed low grade urothelial carcinoma (LGUC) cells (Figure 5F) and was diagnosed as Class IV. Figure 5I showed high grade urothelial carcinoma (HGUC) cells (Figure 5J) and was diagnosed as Class V. These three WSIs should be classified as neoplastic in this study. The heatmap images show true positive predictions of atypical utorhelial cells (Figure 5C,D), LGUC cells (Figure 5G,H), and HGUC cells (Figure 5K,L) which were confirmed by a cytoscreener and a cytopathologist by viewing original WSIs and predicted heatmap images. In contrast, in low probability tiles (light blue and blue background) (Figure 5), two independent cytoscreeners confirmed there were no neoplastic urothelial epithelial cells.

### 3.4. True Negative Prediction

The model ENB1-UC-FS+WS satisfactorily predicted negative cases (cytopathologically as Class I and Class II) in urine LBC WSIs (Figure 6). The heatmap images show true negative predictions of neoplastic urothelial epithelial cells (Figure 6C,F). In zero probability tiles (blue background color) (Figure 6C,F), there are no neoplastic urothelial epithelial cells in pyuria (cytodiagnosed as Class I) (Figure 6A) which consisted of infective fluid with small number of non-neoplastic epithelial cells (Figure 6B) and urothelial epithelial cells with slight nuclear enlargement (Figure 6D,E) (cytodiagnosed as Class II).

### 3.5. False Positive Prediction

A cytopathologically diagnosed negative (Class I) case (Figure 7A) consisted of metaplastic squamous epithelial cells and non-neoplastic urothelial epithelial cells (Figure 7B) was false positively predicted for neoplastic urothelial epithelial cells by our model [ENB1-UC-FS+WS ]. The heatmap image (Figure 7C) shows false positive predictions of neoplastic urothelial epithelial cells with high probability tiles (Figure 7D). Cytopathologically, there are non-neoplastic urothelial epithelial cells with a slightly increased nuclear cytoplasmic (N/C) ratio and metaplastic squamous epithelial cells (Figure 7B), which could be a major cause of false positive.

### 3.6. False Negative Prediction

According to the cytodiagnosis report and additional cytoscreener and cytopathologist’s review, in this urine LBC WSI (Figure 8A), there were cellular clusters of atypical (neoplastic) urothelial epithelial cells (Figure 8B,C) with high nuclear cytoplasmic ratio, indicating this WSI (Figure 8A) should be classified as neoplastic (Class III). However, our model [ENB1-UC-FS+WS ] did not predict or very low level predicted neoplastic urothelial epithelial cells (Figure 8D–F). It would be speculated that neoplastic urothelial epithelial cellular clustering could be a possible cause of false negative due to the overlapping morphology.

## 4. Discussion

In this study, we trained deep learning models for the classification of neoplastic urothelial epithelial cells in WSIs of urine LBC specimens. The best model (ENB1-UC-FS+WS) achieved overall a good performance with ROC-AUCs of 0.984 (CI: 0.969–0.995) on equal balance and 0.990 (CI: 0.982–0.996) on clinical balance test sets and low log-loss values of 0.180 (CI: 0.123–0.259) on equal balance and 0.223 (CI: 0.181–0.284) on clinical balance test sets, which reflects the clinical setting based on the statistics on cytodiagnosis in 2016 to 2021 by the Japanese Society of Clinical Cytology (https://jscc.or.jp/ accessed on 13 January 2022). Both equal and clinical balance test sets (Table 1) were collected based on the cytodiagnoses, reviewed by two independent cytoscreeners or cytopathologists, then verified by a senior cytoscreener or cytopathologist. We ensured that we had consensus on the diagnoses of the test set WSIs. Our best model (ENB1-UC-FS+WS) also achieved high accuracy (0.945–0.946), sensitivity (0.940–0.960), and specificity (0.929–0.946) in WSI level. It has been reported that at urine LBC (ThinPrep) WSI level, the deep learning model predicted neoplastic (positive) WSI at 0.842 (accuracy), 0.795 (sensitivity), and 0.845 (specificity) [57]. Our latest reported uterine cervix LBC (ThinPrep) model demonstrated accuracy at 0.907, sensitivity at 0.850, and specificity at 0.911 at WSI level [35]. In this study, we have trained total four deep learning models (ENB1-UC-FS+WS, ENB1-UC-WS, ENB1-IN-FS+WS, and ENB1-IN-WS) using two different weight initialisation: ImageNet and pre-trained uterine cervix neoplastic LBC model from a previous study [35] (Figure 2). At WSI level, these four models showed almost comparable ROC-AUC, log-loss, accuracy, sensitivity, and specificity (Table 4 and Figure 3). However, there was wide variety of tile level prediction as visualized by the heatmap images between the four models (Figure 4). Based on the WSI and tile level (heatmap) evaluations, we have concluded that the model (ENB1-UC-FS+WS) which was trained using the pre-trained uterine cervix LBC model [35] weight initialisation, performed best. As for the false-negative prediction outputs in the urine LBC WSI which was cytodiagnosed as Class III (Figure 8A), the model (ENB1-UC-FS+WS) could not predict neoplastic atypical urothelial epithelial cell cluster (Figure 8B–F) in which neoplastic urothelial epithelial cells were overlapping and nuclear shapes and structures were hard to determine in the WSI (Figure 8B,C). The model (ENB1-UC-FS+WS) could predict true negative urine crystal and cell debris precisely. False negative prediction outputs were most likely due to neoplastic urothelial epithelial cell clusters that mimicked urine crystal or cell debris.

According to the annual statistics on cytodiagnosis by the Japanese Society of Clinical Cytology (https://jscc.or.jp/, accessed on 13 January 2022), in 2021, there were 2,041,547 urine cytodianosis reports in Japan. In 2021, the total number of cytodiagnosis in Japan was 7,157,413. Therefore, the population of urine cytodiagnosis was approximately 28.5%. In Japan, urine cytology was the second most common cytology in 2021, as cervical cytology was the most common (3,289,877 cases, 50.0%) (https://jscc.or.jp/, accessed on 13 January 2022). LBC of urine specimens is commonly used in cytology laboratories throughout the world and various processing methods, such as ThinPrep and SurePath, have been reported [58,59]. The LBC technique preserves the cells of interest (e.g., urothelial epithelial cells) in a liquid medium and removes most of the debris, blood, and exudate either by filtering or density gradient centrifugation. The efficiency of diagnosis employing the LBC is high because of the cell collection rate. It was demonstrated that the accuracy of diagnoses made employing the LBC method can be increased by understanding the characteristics of the urothelial epithelial cell morphology in suspicious cases [25]. Following the appropriate LBC specimen preparation steps, cell morphology (structure) is satisfactorily preserved, which allows more accurate diagnosing of LBC slides as shown by the significant concordance between cytological and histological diagnosis (92%), the significant number of LGUC (20.5%) revealed by urinary cytology and validated by histology, and the low rate (8%) of misjudgement of cytological diagnosis [60]. In addition, the leftover urine LBC material can be used for other techniques such as immunocytochemistry, molecular biology and flow cytometry [61]. Therefore, LBC has been applied with good results in urine cytology and can be regarded as an appropriate substitute for conventional smear urine cytology. LBC techniques opens new possibilities for a systemic urothelial carcinoma screening by integrating digital pathology WSI technique and deep learning model(s), resulting a standardised high-quality readout (e.g., classification).

One limitation of this study is that it primarily included urine LBC (SurePath) WSIs (both training and test sets) from a single private clinical laboratory in Japan. Therefore, the deep learning models could potentially be biased to such specimens. Validations on a wide variety of specimens from multiple different origins (both clinical laboratories and hospitals) and other LBC method(s) (e.g., ThinPrep) would be essential for ensuring the robustness of the models. Another potential validation study should involve the comparison of the performance of the models against cytoscreeners and cytopathologists in a clinical setting.

## 5. Conclusions

In the present study, we trained deep learning models for the classification of neoplastic urine LBC WSIs. We have evaluated the models on two test sets (equal and clinical balance) achieving ROC-AUCs for diagnosis in the range of 0.984–0.990 by the best model (ENB1-UC-FS+WS). At WSI level, the model (ENB1-UC-FS+WS) achieved high accuracy (0.945–0.946), sensitivity (0.940–0.960), and specificity (0.929–0.946). Not only at WSI level, the model (ENB1-UC-FS+WS) satisfactorily predicted neoplastic urothelial epithelial cells (atypical, LGUC, and HGUC cells) by the heatmap images. Therefore, our model (ENB1-UC-FS+WS) can infer whether the urine LBC WSI is neoplastic (Figure 5) or negative (Figure 6) by inspecting model prediction outputs easily at WSI level as well as heatmap image, which makes it possible to use a deep learning model such as ours as a tool to aid in the urine LBC screening process in the clinical setting (workflow) for ranking cases by order of priority. Cytoscreeners and/or cytopathologists will need to perform full screening and subclassification (e.g., negative, atypical cells, suspicious for malignancy, and malignant) after the primary screening by our deep learning model, which could reduce their working time as the model would have highlighted the suspected neoplastic regions, and they would not have to perform an exhaustive search throughout the entire WSI.

## Figures and Tables

**Figure 1 cancers-15-00226-f001:**
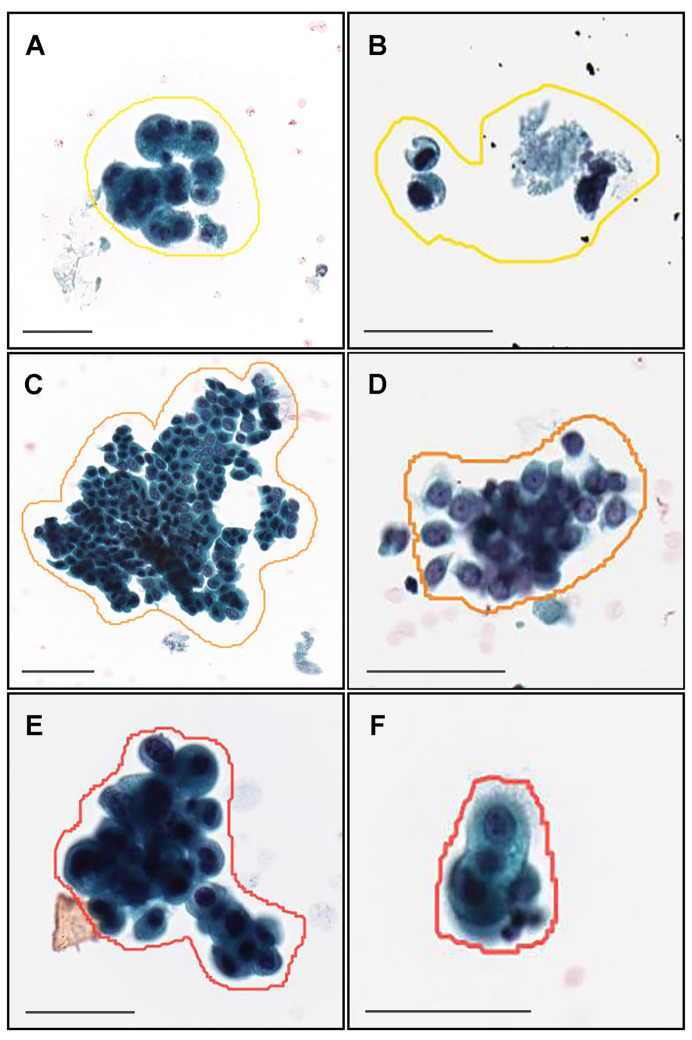
Representative manually drawing annotation image for neoplastic labels on urine liquid-based cytology (LBC) whole slide images (WSIs). The atypical urothelial cells (**A**,**B**) were annotated as atypical cell label. The suspected low grade urothelial carcinoma (LGUC) cells (**C**,**D**) were annotated as LGUC cell label and high grade utorhelial carcinoma (HGUC) cells (**E**,**F**) were annotated as HGUC cell label. The three labels (atypical cell, LGUC cell, and HGUC cell) were grouped as neoplastic label for fully supervised learning. Scale bars are 50 
μ
m.

**Figure 2 cancers-15-00226-f002:**
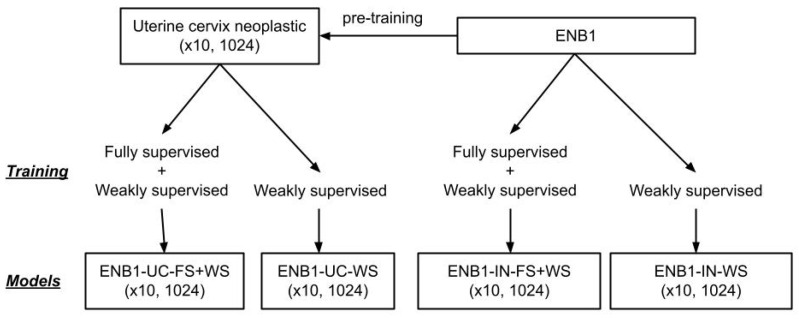
Training method and deep learning models overview. We performed training using two different weight initialisations: ImageNet (IN) and pre-training on a uterine cervix (UC) neoplastic (×10, 1024) dataset from a previous study. We used two different approaches for training: fully supervised (FS) and weakly supervised (WS) learning. This resulted in a total of four models, all trained at magnification ×10 and tile size 1024 × 1024px: ENB1-UC-FS+WS, ENB1-UC-WS, ENB1-IN-FS+WS, and ENB1-IN-WS.

**Figure 3 cancers-15-00226-f003:**
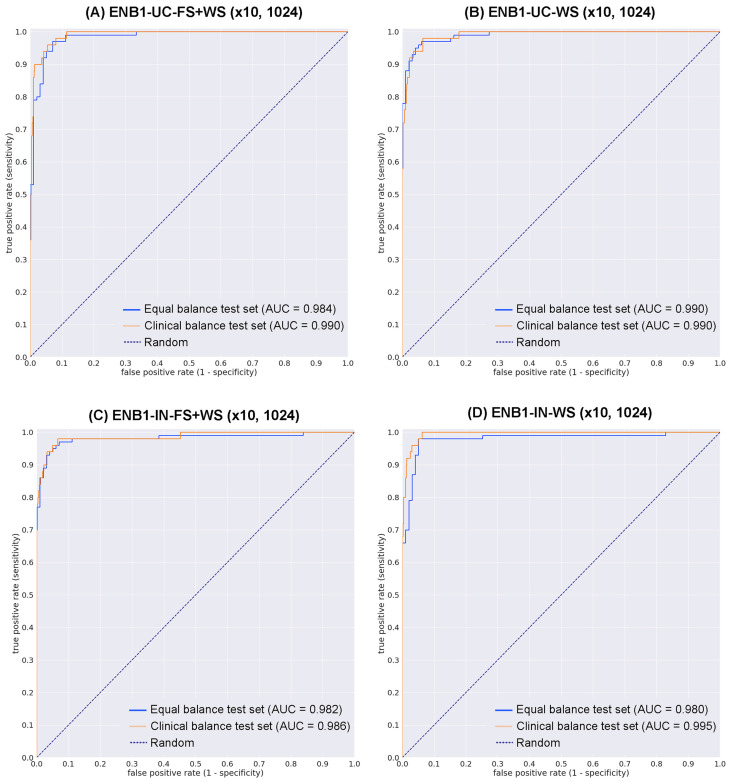
ROC curves on the test sets. (**A**) transfer learning (TL) from uterine cervix liquid-based cytology (LBC) model and fully and weakly supervised learning model, magnification at ×10 and tile size at 1024 × 1024 px (ENB1-UC-FS+WS (×10, 1024)); (**B**) TL from uterine cervix LBC model and weakly supervised learning model, magnification at ×10 and tile size at 1024 × 1024 px (ENB1-UC-WS (×10, 1024)); (**C**) EfficientNetB1 based fully and weakly supervised learning model, magnification at ×10 and tile size at 1024 × 1024 px (ENB1-IN-FS+WS (×10, 1024)); (**D**) EfficientNetB1 based weakly supervised learning model, magnification at ×10 and tile size at 1024 × 1024 px (ENB1-IN-WS (×10, 1024)).

**Figure 4 cancers-15-00226-f004:**
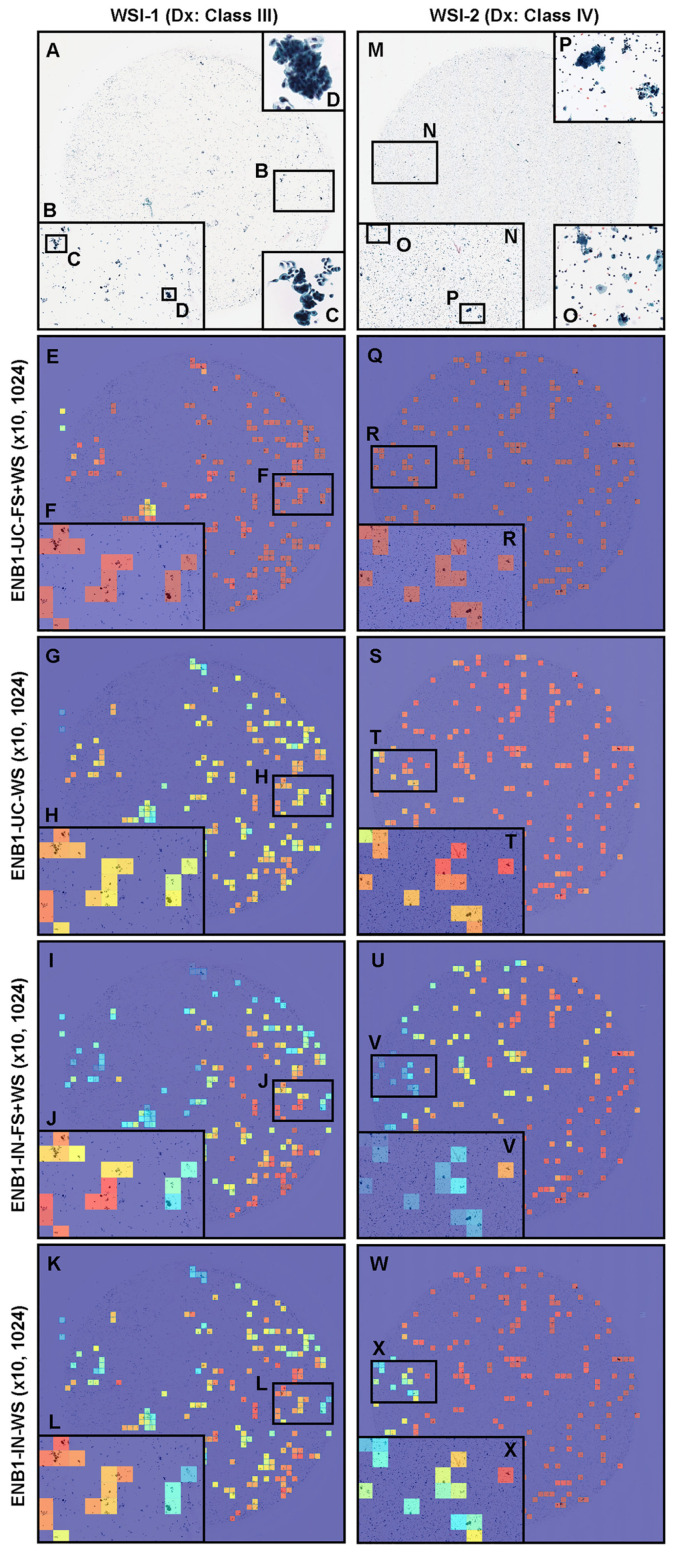
Neoplastic prediction comparison. Comparison of neoplastic predictions in the representative two neoplastic urine liquid-based cytology (LBC) whole-slide images (WSIs) (WSI-1 and WSI-2) of four trained deep learning models (ENB1-UC-FS+WS, ENB1-UC-WS, ENB1-IN-FS+WS, and ENB1-IN-WS). According to the cytopathological diagnostic (Dx) reports, WSI-1 (**A**–**L**) was diagnosed as Class III and WSI-2 (**M**–**X**) was diagnosed as Class IV—both were classified in the neoplastic class in this study. (**A**–**D**,**M**–**P**): LBC cytopathological images for WSI-1 (**A**–**D**) and WSI-2 (**M**–**O**); heatmap prediction images for ENB1-UC-FS+WS model in WSI-1 (**E**,**F**) and WSI-2 (**Q**,**R**); heatmap prediction images for ENB1-UC-WS model in WSI-1 (**G**,**H**) and WSI-2 (**S**,**T**); heatmap prediction images for ENB1-IN-FS+WS model in WSI-1 (**I**,**J**) and WSI-2 (**U**,**V**); heatmap prediction images for ENB1-IN-WS model in WSI-1 (**K**,**L**) and WSI-2 (**W**,**X**). The localization of predicted tiles in neoplastic WSIs (WSI-1 and WSI-2) were almost same in four models (ENB1-UC-FS+WS, ENB1-UC-WS, ENB1-IN-FS+WS, and ENB1-IN-WS). However, the model pre-trained from uterine cervix LBC model with fully and weakly supervised learning (ENB1-UC-FS+WS) showed the highest neoplastic probabilities (**F**,**R**) in neoplastic tiles (**B**–**D**,**N**–**P**) as compared to other models (**G**–**L**,**S**–**X**). The heatmap uses the jet color map where blue indicates low probability and red indicates high probability.

**Figure 5 cancers-15-00226-f005:**
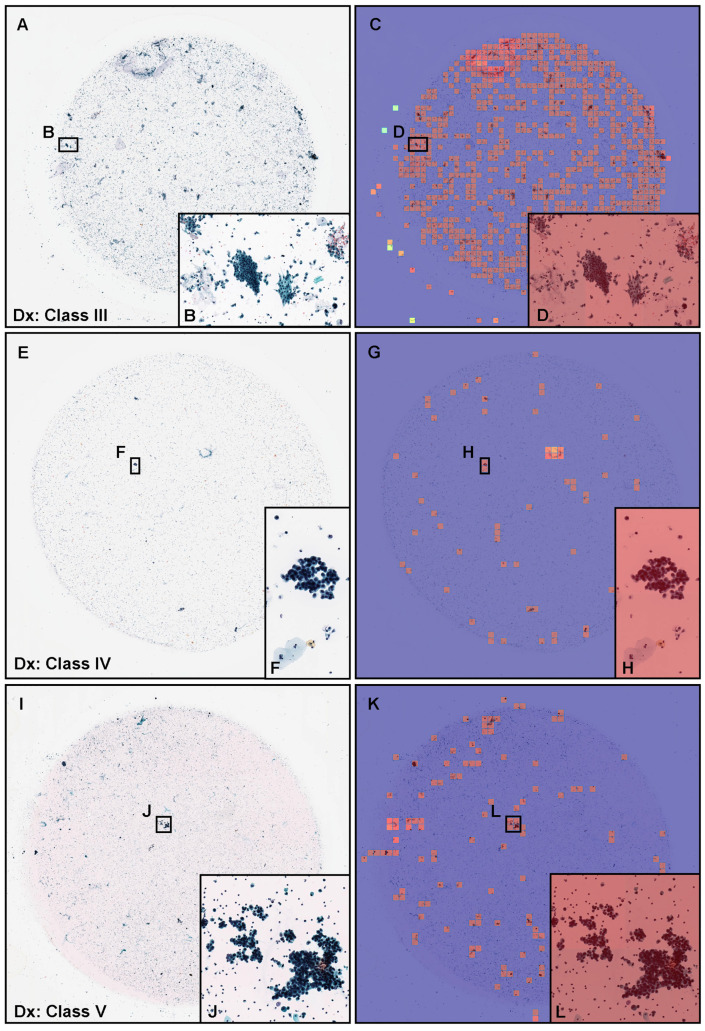
Representative examples of true positive prediction. Neoplastic true positive prediction outputs on urine liquid-based cytology (LBC) whole-slide images (WSIs) from test sets using the ENB1-UC-FS+WS model. According to the cytopathological diagnostic (Dx) reports, (**A**) was diagnosed as Class III with atypical urothelial epithelial cells (**B**), (**E**) was diagnosed as Class IV with suspected low grade urothelial carcinoma (LGUC) cells (**F**), and (**I**) was diagnosed as Class V with suspected high grade utorhelial carcinoma (HGUC) cells (**J**). The heatmap images (**C**,**D**,**G**,**H**,**K**,**L**) show true positive predictions of neoplastic urothelial epithelial cells (**D**,**H**,**L**), which correspond, respectively, to atypical (**B**), suspected LGUC (**F**), and HGUC (**J**) cells. The heatmap uses the jet color map where blue indicates low probability and red indicates high probability.

**Figure 6 cancers-15-00226-f006:**
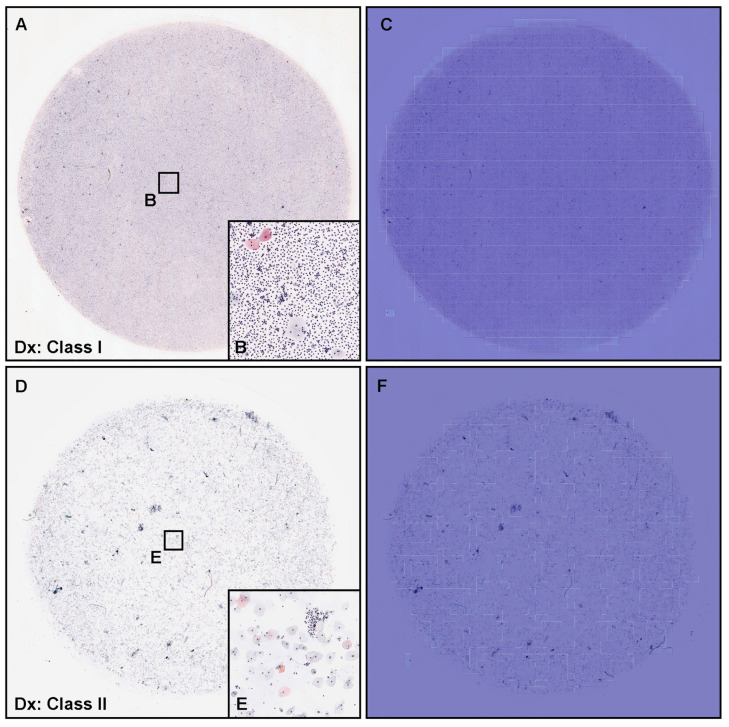
Representative examples of true negative prediction. Two representative examples of neoplastic true negative prediction outputs on urine liquid-based cytology (LBC) whole-slide images (WSIs) from test sets using ENB1-UC-FS+WS model. According to the cytopathological diagnostic (Dx) reports, (**A**) was diagnosed as Class I and (**B**) was Class II, which were negative for urothelial neoplastic epithelial cells. Cytopathologically, (**A**) was pyuria which consisted of infective fluid (pus) with small number of non-atypical epithelial cells (**B**). (**D**,**E**) included urothelial epithelial cells with slight nuclear enlargement. The heatmap images (**C**,**F**) show true negative prediction of neoplastic epithelial cells. The heatmap uses the jet color map where blue indicates low probability and red indicates high probability.

**Figure 7 cancers-15-00226-f007:**
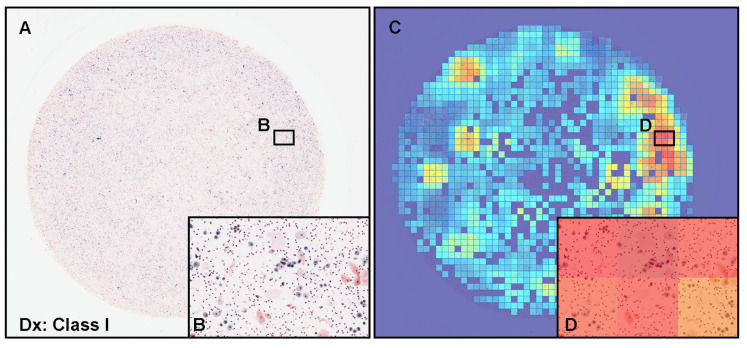
Representative example of false positive prediction. A representative example of neoplastic false positive prediction outputs on urine liquid-based cytology (LBC) whole-slide images (WSIs) from test sets using the ENB1-UC-FS+WS model. According to the cytopathological diagnostic (Dx) report, (**A**) was diagnosed as Class I and consisted of metaplastic squamous epithelial cells and non-atypical (non-neoplastic) urothelial epithelial cells with inflammatory cells (**B**). The heatmap images (**C**,**D**) show false positive predictions (**D**) which correspond, respectively, to (**B**). The heatmap uses the jet color map where blue indicates low probability and red indicates high probability.

**Figure 8 cancers-15-00226-f008:**
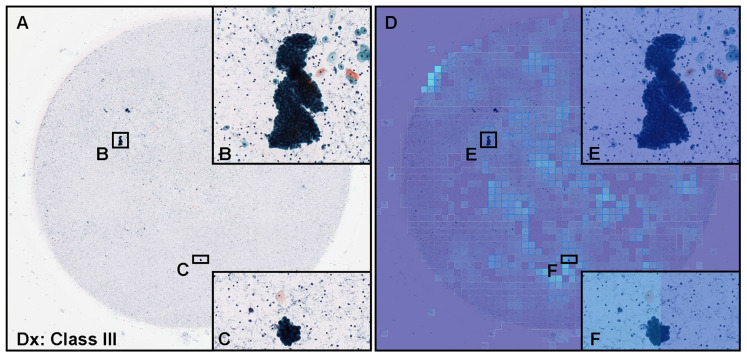
Representative example of false negative prediction. A representative example of neoplastic false negative prediction outputs on urine liquid-based cytology (LBC) whole-slide images (WSIs) from test sets using the ENB1-UC-FS+WS model. According to the cytopathological diagnostic (Dx) report, (**A**) was diagnosed as Class III and included clusters of atypical urothelial epithelial cells (**B**,**C**). The heatmap images (**D**–**F**) show false negative predictions (**E**,**F**) which correspond, respectively, to (**B**,**C**). The heatmap uses the jet color map where blue indicates low probability and red indicates high probability.

**Table 1 cancers-15-00226-t001:** Datasets.

	Training	Validation	Test (Equal Balance)	Test (Clinical Balance)	Total
Negative	724	10	100	500	1334
*Class I*	360	5	50	250	665
*Class II*	364	5	50	250	669
Neoplastic	62	10	100	50	222
*Class III*	38	4	48	20	110
*Class IV*	11	3	23	14	51
*Class V*	13	3	29	16	61
Total	786	20	200	550	1556

**Table 2 cancers-15-00226-t002:** Annotation labels and numbers of annotation.

Annotation Label	Number of Annotation
Atypical cell	9950
Low-grade urothelial carcinoma (LGUC) cell	1646
High-grade urothelial carcinoma (HGUC) cell	1611
Total	13,207

**Table 3 cancers-15-00226-t003:** ROC-AUC and log-loss scores for existing deep learning models to classify liquid-based cytology (LBC) and histopathology whole slide images (WSIs).

Existing Models	ROC-AUC	Log Loss
*Liquid-based cytology (LBC)*		
Uterine cervix Neoplastic (×10, 1024)	0.836 [0.775–0.885]	0.778 [0.620–0.989]

**Table 4 cancers-15-00226-t004:** ROC AUC, log loss, accuracy, sensitivity, and specificity results on the equal balance and clinical balance test sets.

	Test Set	
	Equal Balance	Clinical Balance
ENB1-UC-FS+WS (×10, 1024)		
ROC-AUC	0.984 [0.969–0.995]	0.990 [0.982–0.996]
Log-loss	0.180 [0.123–0.259]	0.223 [0.181–0.284]
Accuracy	0.945 [0.905–0.970]	0.946 [0.924–0.962]
Sensitivity	0.960 [0.920–0.990]	0.940 [0.861–1.000]
Specificity	0.929 [0.862–0.972]	0.946 [0.924–0.964]
ENB1-UC-WS (×10, 1024)		
ROC-AUC	0.990 [0.985–0.999]	0.990 [0.981–0.997]
Log-loss	0.251 [0.178–0.295]	0.098 [0.081–0.119]
Accuracy	0.955 [0.935–0.985]	0.940 [0.920–0.960]
Sensitivity	0.950 [0.911–0.990]	0.980 [0.933–1.000]
Specificity	0.960 [0.931–1.000]	0.936 [0.915–0.958]
ENB1-IN-FS+WS (×10, 1024)		
ROC-AUC	0.982 [0.957–0.996]	0.986 [0.963–0.998]
Log-loss	0.225 [0.156–0.321]	0.082 [0.063–0.106]
Accuracy	0.950 [0.910–0.975]	0.936 [0.918–0.956]
Sensitivity	0.930 [0.863–0.971]	0.960 [0.894–1.000]
Specificity	0.970 [0.930–1.000]	0.934 [0.914–0.955]
ENB1-IN-WS (×10, 1024)		
ROC-AUC	0.980 [0.960–0.997]	0.995 [0.990–0.998]
Log-loss	0.258 [0.185–0.289]	0.128 [0.114–0.144]
Accuracy	0.960 [0.940–0.990]	0.944 [0.924–0.960]
Sensitivity	0.970 [0.945–1.000]	1.000 [1.000–1.000]
Specificity	0.950 [0.914–0.990]	0.938 [0.915–0.956]
ResNet50V2-IN-FS+WS (×10, 1024)		
ROC-AUC	0.962 [0.919–0.986]	0.972 [0.935–1.000]
Log-loss	0.238 [0.145–0.357]	0.085 [0.050–0.124]
Accuracy	0.916 [0.865–0.955]	0.915 [0.884–0.950]
Sensitivity	0.888 [0.812–0.937]	0.949 [0.874–1.000]
Specificity	0.945 [0.895–0.993]	0.914 [0.875–0.950]
DenseNet121-IN-FS+WS (×10, 1024)		
ROC-AUC	0.945 [0.905–0.977]	0.957 [0.922–0.988]
Log-loss	0.233 [0.152–0.345]	0.185 [0.146–0.224]
Accuracy	0.919 [0.867–0.962]	0.925 [0.887–0.958]
Sensitivity	0.919 [0.835–0.977]	0.921 [0.846–0.971]
Specificity	0.957 [0.905–1.000]	0.906 [0.869–0.945]
InceptionV3-IN-FS+WS (×10, 1024)		
ROC-AUC	0.959 [0.923–0.983]	0.978 [0.940–1.000]
Log-loss	0.239 [0.151–0.354]	0.186 [0.177–0.198]
Accuracy	0.912 [0.857–0.955]	0.924 [0.895–0.959]
Sensitivity	0.898 [0.820–0.957]	0.956 [0.878–1.000]
Specificity	0.954 [0.895–0.995]	0.906 [0.868–0.941]

## Data Availability

The datasets generated during and/or analysed during the current study are not publicly available due to specific institutional requirements governing privacy protection; however, they are available from the corresponding author and from the private clinical laboratory in Japan on reasonable request. The datasets that support the findings of this study are available from the private clinical laboratory (Japan), but restrictions apply to the availability of these data, which were used under a data-use agreement that was made according to the Ethical Guidelines for Medical and Health Research Involving Human Subjects as set by the Japanese Ministry of Health, Labour and Welfare (Tokyo, Japan) and, thus, are not publicly available. However, the datasets are available from the authors upon reasonable request for private viewing and with permission from the corresponding private clinical laboratory within the terms of the data use agreement and if compliant with the ethical and legal requirements as stipulated by the Japanese Ministry of Health, Labour and Welfare.

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
