# Peer review of "Deep Learning-Based Screening of Urothelial Carcinoma in Whole Slide Images of Liquid-Based Cytology Urine Specimens"

_cancers, 2022, doi:10.3390/cancers15010226_

Round 1

Reviewer 1 Report (Previous Reviewer 1)

In this paper, the authors propose to use deep learning to sort urine LBC whole-slide images (WSIs) into neoplastic and non-neoplastic (negative) groups.
Thank you for the efforts in comparing this version with the previous version;
especially the use and comparison with other deep learning models like ResNet, etc...
My remarks to improve the presentation of your paper are:
1/ Regarding your introduction, add a paragraph at the end for the distribution of your paper.
2/ Include a subsection 3.1 evaluation metric that describes the evaluation metrics, including equations for accuracy, roc, and so on.
3/ Include a table similar to related work to compare existing works.
I wish you a good continuation.

Author Response

Reviewer 1:

In this paper, the authors propose to use deep learning to sort urine LBC whole-slide images (WSIs) into neoplastic and non-neoplastic (negative) groups.

Thank you for the efforts in comparing this version with the previous version;

especially the use and comparison with other deep learning models like ResNet, etc...

My remarks to improve the presentation of your paper are:

1/ Regarding your introduction, add a paragraph at the end for the distribution of your paper.

Response: With distribution do you mean a summary of the sections to follow?

2/ Include a subsection 3.1 evaluation metric that describes the evaluation metrics, including equations for accuracy, roc, and so on.

Response: We have had a similar thing in a previous paper but were told to remove them as they are common and ubiquitous metrics, given that everyone doing any sort of data analysis would be familiar with or would be easily able to look up.

3/ Include a table similar to related work to compare existing works.

I wish you a good continuation.

Response: As None of the other related works use the same dataset, it would be impossible to perform a direct comparison of metrics with other works.

Reviewer 2 Report (Previous Reviewer 3)

My previous recommendation was "accept after minor corrections", now as the authors responded to the comments and improved their manuscript appropriately I would recommend accepting in the present form.

Thanks

Author Response

Reviewer 2:

My previous recommendation was "accept after minor corrections", now as the authors responded to the comments and improved their manuscript appropriately I would recommend accepting in the present form.

Response: Thank you.

Reviewer 3 Report (Previous Reviewer 2)

Considering the high incidence of bladder and excretory tract tumors, the importance of cytological examination is undeniable. In this manuscript, the authors emphasize that it is important to integrate new deep learning methods that can automatically and rapidly diagnose a large amount of specimens without delay. Compared to the previous version, this new form of the manuscript is more complete and detailed in every point.

Author Response

Reviewer 3:

Considering the high incidence of bladder and excretory tract tumors, the importance of cytological examination is undeniable. In this manuscript, the authors emphasize that it is important to integrate new deep learning methods that can automatically and rapidly diagnose a large amount of specimens without delay. Compared to the previous version, this new form of the manuscript is more complete and detailed in every point.

Response: Thank you.

This manuscript is a resubmission of an earlier submission. The following is a list of the peer review reports and author responses from that submission.

Round 1

Reviewer 1 Report

In this paper, the authors propose to use deep learning to classify the urine LBC whole-slide images (WSIs) into neoplastic and non-neoplastic (negative). I have some remarks to make to improve this version and some questions:

At the presentation level:

1 / I find that you put a page and a half into describing your database but only one page into clarifying your model, and I find that you can improve the description of your model even with a diagram or an algorithm.

2) About your introduction, I cannot find your contribution exactly, and you can add a paragraph at the end of the introduction for the distribution of your paper.

3) the figures and results; even if you used latex, try to place the figures at the appropriate paragraph level.

4) The titles of the figures must be symbolic in the form of a few words and not a paragraph of more than 7 lines.

At the scientific level:

1) You used a dataset of 750 examples, and I find that this one is very small for deep learning training, and you can use a simple machine learning classifier, and I'm sure you will find good results too;

2) Try to use the dta-augmentation technique to increase your dataset and describe the number of new ones.

3) You've used the efficientNet deep learning model, but you haven't compared it to other models, even though you know there are many models for a simple binary classification task;

4) I hope you improve your related works, especially by adding your work to other deep learning models and comparing it to them.

5) What is the difference between this approach and your published approach [29] Other than that, you changed the database from prostate to urine.

6) Improve the presentation of your contribution with more flowcharts or in other ways, and 

Describe the difference between your approach and the other approaches proposed at the level of the medical image classification 

Author Response

Reviewer 1:

In this paper, the authors propose to use deep learning to classify the urine LBC whole-slide images (WSIs) into neoplastic and non-neoplastic (negative). I have some remarks to make to improve this version and some questions:

At the presentation level:

1) I find that you put a page and a half into describing your database but only one page into clarifying your model, and I find that you can improve the description of your model even with a diagram or an algorithm.

Response: We did not develop a new method for training the model and this is not a methodology paper. We used an existing method described in more detail in a previous paper, which we cite.

2) About your introduction, I cannot find your contribution exactly, and you can add a paragraph at the end of the introduction for the distribution of your paper.

Response: We have separated the paragraph that pertains to the contribution at the end of the introduction to make it more clear.

3) the figures and results; even if you used latex, try to place the figures at the appropriate paragraph level.

Response: We have moved the figures around to be close to the text.

4) The titles of the figures must be symbolic in the form of a few words and not a paragraph of more than 7 lines.

Response: We have shortened the first sentences of the figure captions.

At the scientific level:

1) You used a dataset of 750 examples, and I find that this one is very small for deep learning training, and you can use a simple machine learning classifier, and I'm sure you will find good results too;

Response: These are whole slide images that are really large with sizes on average of 60K x 60K pixels. So each whole slide images, once tiled, results in 60000^2/512^2 ~= 13730 tiles, so in  total 750x13730 ~ 10 million 512x512px images.

2) Try to use the dta-augmentation technique to increase your dataset and describe the number of new ones.

Response:  We have used augmentation during training, and we do mention in the methods section that  “..we performed real-time augmentation of the extracted tiles using variations of brightness, saturation, and contrast.”

3) You've used the efficientNet deep learning model, but you haven't compared it to other models, even though you know there are many models for a simple binary classification task;

Response: This is a clinical validation paper and not a methodology paper. The efficient net model was deemed most appropriate for this dataset based on an earlier paper.

4) I hope you improve your related works, especially by adding your work to other deep learning models and comparing it to them.

Response: We do cite other related work. See citations [34-43].

5) What is the difference between this approach and your published approach [29] Other than that, you changed the database from prostate to urine.

Response: That is the main difference.

6) Improve the presentation of your contribution with more flowcharts or in other ways, and describe the difference between your approach and the other approaches proposed at the level of the medical image classification

Response: This is a clinical validation study on a urine cytology dataset and not a methodology paper.

Reviewer 2 Report

This study investigates the use of deep learning models for the classification of urine LBC whole-slide images (WSIs) into neoplastic and non-neoplastic (negative), demonstrating the promising potential use of author's model for aiding urine cytodiagnostic processes. Despite some limitations that could be biased, the study is interesting if we consider that the 48.6% of biopsy proven low-grade urothelial carcinomas have a urine cytodiagnosis of atypical or neoplastic suspicious. For this reason, liquid-based cytology (LBC) was developed as an alternative to conventional smear cytology, considering that the efficiency of diagnosis employing the LBC is high because of the cell collection rate. The study methodology is appropriate and the results agree with the premises. However, the authors are requested to replace the incorrect term "ceuular" with the term "cellular" in section 3.6. False Negative Prediction. Then, in the introduction, the authors write "Most of the bladder cancers are urothelial in origin (approximately 90% of bladder cancers) and primary adenocarcinoma of the bladder is rare [2,3]". In this regard, the authors are advised to evaluate the inclusion, as a bibliographic reference, of the recent paper by Baio et al (Baio R, Spiezia N, Marani C and Schettini M: Potential contribution of benzodiazepine abuse in the development of a bladder sarcomatoid carcinoma: A case report. Mol Clin Oncol 15: 231, 2021) in which the authors underlined that invasive urothelial carcinoma has a propensity for disparate differentiation and presentation of morphological variants.

Author Response

Reviewer 2:

This study investigates the use of deep learning models for the classification of urine LBC whole-slide images (WSIs) into neoplastic and non-neoplastic (negative), demonstrating the promising potential use of author's model for aiding urine cytodiagnostic processes. Despite some limitations that could be biased, the study is interesting if we consider that the 48.6% of biopsy proven low-grade urothelial carcinomas have a urine cytodiagnosis of atypical or neoplastic suspicious. For this reason, liquid-based cytology (LBC) was developed as an alternative to conventional smear cytology, considering that the efficiency of diagnosis employing the LBC is high because of the cell collection rate. The study methodology is appropriate and the results agree with the premises. 

Response: Thank you so much.

However, the authors are requested to replace the incorrect term "ceuular" with the term "cellular" in section 3.6. False Negative Prediction.

Response: Fixed. Thanks.

Then, in the introduction, the authors write "Most of the bladder cancers are urothelial in origin (approximately 90% of bladder cancers) and primary adenocarcinoma of the bladder is rare [2,3]". In this regard, the authors are advised to evaluate the inclusion, as a bibliographic reference, of the recent paper by Baio et al (Baio R, Spiezia N, Marani C and Schettini M: Potential contribution of benzodiazepine abuse in the development of a bladder sarcomatoid carcinoma: A case report. Mol Clin Oncol 15: 231, 2021) in which the authors underlined that invasive urothelial carcinoma has a propensity for disparate differentiation and presentation of morphological variants.

Response:  We have added a reference to that paper.

Reviewer 3 Report

The paper of Tsuneki et al. is about developing and validating a machine-learning approach to diagnose cases with urothelial carcinoma. The topic is interesting and from my experience, I can tell that the methodology is sound, nonetheless, the work was not presented clearly and adequately. The current methodology part cannot be followed by other authors because it lacks some basic and necessary information. Overall, comprehensive improvement is required for this paper before getting published, which in this form, I would recommend major revision.

The followings are some comments that may help the authors improve their paper

·         “Simple summary” is not clear, the authors mentioned that they developed a model, so, what do they mean by "the best model"? this statement provides indications that there are several models while the best model was ..... please clarify this point

·         the abstract is not adequate and does not reflect the study content. Please provide more details about the methodology. Please avoid using vague terms at this stage like "equal and clinical balance" unless appropriate clarification was provided.

·         The term “ROC-AUC” is not usually used by authors, you can simply use "area under the curve".

·         WSI is a technical term not a pathological condition to say "WSI diagnosis". Please use alternative terms in the whole text.

·         Please use “please use "high cellular yields" instead of "high cell collection rate" in the abstract

·         For the introduction, please add a section about the significance of using LBC in other human malignancies to justify the utility of this approach in your project. For this purpose, the authors can cite DOI: 10.1002/cncy.22599 AND doi: 10.1136/bmj.39262.506528.47

·         please add appropriate citations for the first 6 lines in the introduction

·         please add citations for the section about the utility of LBC over conventional smear

·         In the methodology, please define the criteria that were used to consider a specimen as inadequate for exclusion

·         Provide definitions for classes I to IV

·         the majority of information in section 2.2. were already mentioned in section 2.1. please remove duplication

·         what do equal and clinical balance mean?

·         Section 2.3. the first 3 lines are very wordy and not clear. A section like that can be summarized and simplified by "a cohort of 62 training cases and 10 validation cases were manually annotated by experienced pathologists"

·         For the OpenSeadragon tool, the link does go to the drawing tool

·         Section 2.4. please cite “a previous study”

Author Response

Reviewer 3:

The paper of Tsuneki et al. is about developing and validating a machine-learning approach to diagnose cases with urothelial carcinoma. The topic is interesting and from my experience, I can tell that the methodology is sound, nonetheless, the work was not presented clearly and adequately. The current methodology part cannot be followed by other authors because it lacks some basic and necessary information. Overall, comprehensive improvement is required for this paper before getting published, which in this form, I would recommend major revision.

The followings are some comments that may help the authors improve their paper

  •         “Simple summary” is not clear, the authors mentioned that they developed a model, so, what do they mean by "the best model"? this statement provides indications that there are several models while the best model was ..... please clarify this point

Response: We have clarified that we “..trained models using four different approaches..”, one of which performed best.

  •         the abstract is not adequate and does not reflect the study content. Please provide more details about the methodology. Please avoid using vague terms at this stage like "equal and clinical balance" unless appropriate clarification was provided.

Response: We’ve added a clarification for clinical balance that it’s “one of which was representative of the clinical distribution of neoplastic cases”

  •         The term “ROC-AUC” is not usually used by authors, you can simply use "area under the curve".

Response: We’ve replaced it with “area under the curve”.

  •         WSI is a technical term not a pathological condition to say "WSI diagnosis". Please use alternative terms in the whole text.

Response: WSI diagnosis refers to a diagnosis performed by a pathologist using a whole slide image, as opposed to microscopic diagnosis which is one performed via a microscope. It’s a term used in other publications. See for example [1,2,3]. If, however, you still find this to be not clear, we’re happy to change it to a different term.

[1] Mukhopadhyay S, el al. Whole Slide Imaging Versus Microscopy for Primary Diagnosis in Surgical Pathology: A Multicenter Blinded Randomized Noninferiority Study of 1992 Cases (Pivotal Study). Am J Surg Pathol. 2018 Jan;42(1):39-52. doi: 10.1097/PAS.0000000000000948. PMID: 28961557; PMCID: PMC5737464.

“...There was only 1 case in which the WSI diagnosis of 2 (of 4) readers was judged as a major discordance when the corresponding microscopic diagnosis was concordant or minor discordant…”

[2] https://www.frontiersin.org/articles/10.3389/fonc.2022.918580/full

[3] https://bmcclinpathol.biomedcentral.com/articles/10.1186/1472-6890-6-4

“the effect of compression has not been carefully studied on WSI diagnosis and its clinical impact (or lack there of) should not be taken on faith.”

  •         Please use “please use "high cellular yields" instead of "high cell collection rate" in the abstract

Response: Changed.

  •         For the introduction, please add a section about the significance of using LBC in other human malignancies to justify the utility of this approach in your project. For this purpose, the authors can cite DOI: 10.1002/cncy.22599 AND doi: 10.1136/bmj.39262.506528.47

Response: We’ve added a citation for those two.

  •         please add appropriate citations for the first 6 lines in the introduction

Response: We have added the relevant citations.

“For routine clinical practices, clinicians obtain urinary tract cytology specimens for the screening of urothelial carcinoma [Ref: #1, #2]. Urine specimens play a critical role in the clinical evaluation of patients who have clinical signs and symptoms (e.g., haematuria and painful urination) suggestive of pathological changes within the urinary tract [Ref: #3]. Urothelial carcinoma is the most common malignant neoplasm detected by urine cytology [Ref: #2].”

#1: https://pubmed.ncbi.nlm.nih.gov/30375194/

#2: https://pubmed.ncbi.nlm.nih.gov/10696242/

#3: https://pubmed.ncbi.nlm.nih.gov/24617100/

  •         please add citations for the section about the utility of LBC over conventional smear

Response: We have added the relevant citations.

“LBC has several advantages in preparation and diagnostic process compared with conventional smear [Ref: #1, #2, #3, #4]. The LBC technique preserves the cells of interest in a liquid medium and removes most of the debris, blood, and exudate either by filtering or density gradient centrifugation [Ref: #5, #6]. LBC provides automated and standardized processing techniques that produce a uniformly distributed and cell-enriched slide [Ref: #7, #8, #9]. Moreover, residual specimens can be used for additional investigations (e.g., immunocytochemistry) [Ref: #10, #11, #12].”

#1: https://pubmed.ncbi.nlm.nih.gov/25948938/

#2: https://pubmed.ncbi.nlm.nih.gov/35596035/

#3: https://pubmed.ncbi.nlm.nih.gov/35579303/

#4: https://pubmed.ncbi.nlm.nih.gov/20073599/

#5: https://pubmed.ncbi.nlm.nih.gov/11754203/

#6: https://pubmed.ncbi.nlm.nih.gov/9479337/

#7: https://pubmed.ncbi.nlm.nih.gov/35928530/

#8: https://pubmed.ncbi.nlm.nih.gov/32368601/

#9: https://pubmed.ncbi.nlm.nih.gov/23585899/

#10: https://pubmed.ncbi.nlm.nih.gov/35928527/

#11: https://pubmed.ncbi.nlm.nih.gov/35574396/

#12: https://pubmed.ncbi.nlm.nih.gov/26204907/

  •         In the methodology, please define the criteria that were used to consider a specimen as inadequate for exclusion

Response: We’ve added an additional clarification.

  •         Provide definitions for classes I to IV

Response: We’ved added the following:

Class I: negative for HGUC; Class II: negative for HGUC with reactive urothelial epithelial cells; Class III: atypical urothelial epithelial cells and suspicious for LGUC; Class IV: LGUC and suspicious for HGUC; Class V: HGUC

  •         the majority of information in section 2.2. were already mentioned in section 2.1. please remove duplication

Response: We’ve merged the sections and removed duplications.

  •         what do equal and clinical balance mean?

Response: The equal balance refers to a test where the distribution of positive and negative cases were equal, while a clinical balance refers to a set where the distribution of cases is reflective of clinical cases.

  •         Section 2.3. the first 3 lines are very wordy and not clear. A section like that can be summarized and simplified by "a cohort of 62 training cases and 10 validation cases were manually annotated by experienced pathologists"

Response: We’ve changed it as suggested.

  •         For the OpenSeadragon tool, the link does go to the drawing tool

Response: It is the main site for the tool.

  •         Section 2.4. please cite “a previous study”

Response: We’ve cited the previous study

Round 2

Reviewer 1 Report

thank you for all your answers, but I was expecting better answers than "This is a clinical validation study..." especially when I ask to try another deep learning approach other than efficientNet,

If your method is already published and efficintNet has already been proven by other studies... so why we must accept this paper in this journal?

Reviewer 3 Report

The author improved their manuscript and worked on some comments, but further improvement is still required.

·         In the abstract, use “area under the curve” only without ROC.

·         The abstract needs a conclusion part.

·         For LBC, inadequate refers to inadequate cellularity rather than poor scanning quality.

·         I understand that “WSI diagnosis” refers to the diagnosis performed by pathologists based on WSI. The papers that used “WSI diagnosis” were basically comparative studies that used this term to differentiate between WSI and light microscope, rather than refer to the diagnosis by itself. In this paper, the authors can say “each WSI was assessed to xxx to confirm the diagnosis” and that’s it. After that, the authors can only use the term “diagnosis” without WSI diagnosis.